# A Comprehensive Characterization of Pyrolysis Oil from Softwood Barks

**DOI:** 10.3390/polym11091387

**Published:** 2019-08-23

**Authors:** Haoxi Ben, Fengze Wu, Zhihong Wu, Guangting Han, Wei Jiang, Arthur J. Ragauskas

**Affiliations:** 1Key Laboratory of Energy Thermal Conversion and Control of Ministry of Education, Southeast University, Nanjing 210096, China; 2Qingdao University, Qingdao 266071, China; 3State Key Laboratory of Bio-Fibers and Eco-Textiles, Qingdao University, Qingdao 266071, China; 4Joint Institute for Biological Sciences, Biosciences Division, Oak Ridge National Lab, Oak Ridge, TN 37831, USA; 5Department of Chemical and Biomolecular Engineering, University of Tennessee, Knoxville, TN 37996, USA

**Keywords:** pyrolysis oils, biochar, elemental analysis, nuclear magnetic resonance (NMR) analysis

## Abstract

Pyrolysis of raw pine bark, pine, and Douglas-Fir bark was examined. The pyrolysis oil yields of raw pine bark, pine, and Douglas-Fir bark at 500 °C were 29.18%, 26.67%, and 26.65%, respectively. Both energy densification ratios (1.32–1.56) and energy yields (48.40–54.31%) of char are higher than pyrolysis oils (energy densification ratios: 1.13–1.19, energy yields: 30.16–34.42%). The pyrolysis oils have higher heating values (~25 MJ/kg) than bio-oils (~20 MJ/kg) from wood and agricultural residues, and the higher heating values of char (~31 MJ/kg) are comparable to that of many commercial coals. The elemental analysis indicated that the lower O/C value and higher H/C value represent a more valuable source of energy for pyrolysis oils than biomass. The nuclear magnetic resonance results demonstrated that the most abundant hydroxyl groups of pyrolysis oil are aliphatic OH groups, catechol, guaiacol, and *p*-hydroxy-phenyl OH groups. The aliphatic OH groups are mainly derived from the cleavage of cellulose glycosidic bonds, while the catechol, guaiacol, and *p*-hydroxy-phenyl OH groups are mostly attributed to the cleavage of the lignin β–O-4 bond. Significant amount of aromatic carbon (~40%) in pyrolysis oils is obtained from tannin and lignin components and the aromatic C–O bonds may be formed by a radical reaction between the aromatic and aliphatic hydroxyl groups. In this study, a comprehensive analytical method was developed to fully understand and evaluate the pyrolysis products produced from softwood barks, which could offer valuable information on the pyrolysis mechanism of biomass and promote better utilization of pyrolysis products.

## 1. Introduction

Increasing world energy consumption [1] and growing carbon dioxide emissions have contributed to the promotion of sustainable energy development. Biomass is considered a sustainable raw material for the manufacture of chemicals and fuels [2,3,4] because it is carbon neutral, quite abundant, renewable, and no competition with food [5,6]. According to related research [7,8,9,10,11,12], various chemicals such as regenerated microfibrillated cellulose fibers and carboxycellulose nanofibers have been successfully produced from jute fibers and spinifex grass. As the most promising fuel raw material, bark is a waste produced by forestry-related industries. Around 30 million tons of bark are generated in the United States and half of them are burnt to generate energy, but the rest will be sent to the landfills [13]. Compare to the gasification and biochemical conversion of biomass, pyrolysis is a development technology to convert biomass into bio-oil and biochar via heating at 400–600  °C in an anoxic atmosphere [14,15,16], which has been reported as the most economical way of bio-fuel transformation [17] Several researchers have accomplished some experiments of pyrolysis of waste biomass. For examples, Yassir et al. [18] rapidly pyrolyzed palm waste at 525 °C with an oil yield of 38.8%, but the higher heating value of these bio-oils is 20.88 MJ/kg, which is much lower than that of bio-oil from softwood barks. Pinto et al. [19] who fast pyrolyzed tannins from pine bark at 400–600 °C have found that the maximum bio-oil yield (37 wt %) measured on a laboratory scale is at 550  °C. In addition, the catechol and 4-methylcatechol, which have a highest yield of 4.4% and 2.3%, respectively, are also coming from the pyrolysis of tannins at 550 °C. Cardoso et al. [20] pyrolyzed bark of bocaiuva residues with the granulometry of 3.0 mm, nitrogen flow of 1 mL/min, mass of 10 g, heating rate of 100 °C/min. Their research showed that the highest outputs of bio-oil at 500, 600, and 700 °C are 25.3%, 22.6%, and 20.9%, respectively. Moreover, Ranjeet et al. [21] proposed that the maximum liquid yield was achieved because of the higher heat transfer and mass transfer between the particles at 500 °C. They also indicated that at higher treatment temperatures, the secondary reactions will occur which lead to a decreasing yield of pyrolysis oil.

For analytical methods, most previous pyrolysis studies used either gas chromatography–mass spectrometry (GC–MS) or Fourier transform infrared spectroscopy (FTIR) or a combination of FTIR and GC–MS for characterization. Sakthivel et al. [22] pyrolyzed the bark of calophyllum inophyllum at 550 °C and characterized the pyrolysis oils with FTIR. Strong signals of functional groups such as C=C, –CH, C=O, –OH groups were observed, but their contents were not quantitative analyzed. Shao et al. [23] used TG–FTIR to analyze pyrolysis products of larch bark at 10–50 °C/min. It can be observed that the pyrolysis of larch bark begins at about 180 °C and most different types of pyrolysis gases and various functional groups are generated within the range of 180–530 °C. Schellekens et al. [24] slowly pyrolyzed different agricultural residues at 350, 450, and 750 °C by using pyrolysis–GC/MS to investigate the molecular properties of pyrolysis products. They found that the pyrolysis temperature is the dominant reason for determining the composition of pyrolysis char regardless of the pyrolysis materials. As the pyrolysis temperature increased, the high molecular weight products of all chemical groups of biochar diminished. However, bio-oil has very complex compositions, containing hundreds of chemical compounds and a substantial amount of water [25,26]. In order to analyze pyrolysis oils more accurately and solve the limitations of FTIR and GC–MS, in this study, the analysis of the chemical structures of pyrolysis oil obtained from pine and Douglas-Fir bark will be accomplished by nuclear magnetic resonance (NMR), which is an alternative and more powerful method to characterize the components in the pyrolysis oils [27,28,29,30].

To provide more detailed information about pyrolysis products from one of the major waste biomass in the US—barks, several precise characterization methods were employed, including quantitative ^31^P NMR, which can provide quantitative information on all the OH groups in the pyrolysis oil and these functional groups normally respond to the aging, acidity, stability, solubility and other problems of the pyrolysis oil; quantitative ^13^C NMR which can provide very valuable information on the major functional groups and can facilitate the identification for the products from both lignin and cellulose parts of bark; Gel Permeation Chromatography (GPC) and elemental analysis can support the conclusions made from the NMR analysis and provide more target information on the whole portion of pyrolysis oil. The mass and energy yield are also very important for the possible industrial applications of pyrolysis oil. Therefore, in this research, valuable information and conclusions can be obtained by performing these characterizations on the products from waste biomass—bark, and they will also facilitate the further industrial applications of pyrolysis products. The concepts established in this research will develop a new way to evaluate and utilize the pyrolysis products by combining several quantitative methods to investigate the pyrolysis oil produced from softwood barks.

## 2. Materials and Methods

### 2.1. Material Preparation

All reagents involved in the research were acquired from VWR International and utilized in accordance with relevant regulations. Douglas-Fir bark was obtained from a local kraft mill in Washington State. Raw Loblolly pine bark is a mixture of wood (~25 wt %) and bark (~75 wt %) purchased from a local kraft pulp mill in Georgia. The pure Loblolly pine bark was collected by manually separating from raw Loblolly pine bark and washed with water. The dried (50 °C, 48 h in a convection oven) samples were referred as pine bark. All samples were ground by Wiley mill and selected through a 0.50 mm screen. The screened barks were dried at 50 °C for 48 h under high vacuum and stored at ~0 °C before use.

### 2.2. Procedure of Pyrolysis Barks

Pyrolysis of various softwood barks was carried out in tubular furnaces. First, a quartz boat with 4.00 g biomass was moved to a position closed to the center of pyrolysis tube. Then, the nitrogen tube was inserted into the left side of the pyrolysis tube, and the right side of the pyrolysis tube was connected to a condenser immersed in an ethanol solution at −10 °C. Prior to heating, the pyrolysis tube was flushed with nitrogen to maintain a nitrogen atmosphere within the pyrolysis tube. In the pyrolysis process, the heating rate measured using a K-type thermocouple was ~2.7 °C/s, and the pyrolysis oils were discharged through condenser. After the pyrolysis was completed, the pyrolysis tube was taken out of the split-tube furnace and constant nitrogen was still kept flowing into the pyrolysis tube until it was cooled to room temperature. Finally, the nitrogen tube and condenser units were removed. The pyrolysis oils, usually composed of two immiscible phases (light oil and heavy oil), were collected by washing the pyrolysis tube and condenser with acetone and then evaporating under reduced pressure. The obtained pyrolysis char and oil were used for subsequent chemical analysis and the char and oil yields were calculated by the gravimetric method.

### 2.3. Measurement of Higher Heating Value (HHV)

The HHVs of the pyrolysis products were determined by the literature methods [31,32] on a Parr 1261 isoperibol bomb calorimeter. The HHVs presented are based on the average of repeated trials with an error of 0.6%.

### 2.4. Measurement of Molecular Weights

According to the methods described in the literature [27,28,30], the number average molecular weight (M*_n_*) and the weight average molecular weight (M*_w_*) of pyrolysis oil were separately measured by GPC analysis. Firstly, the pyrolysis oil sample was mixed with tetrahydrofuran (1 mg/mL). After the sample was sufficiently dissolved, it was filtered using a 0.45 µm syringe filter. Then, the sample was injected with a micro syringe into the Polymer Standard Service Security 1200 system, which was equipped with Agilent High Performance Liquid Chromatography vacuum degasser, ultraviolet detector (270 nm), refractive index detector, and isocratic pump. Four Waters Styragel columns (HR0.5, HR2, HR4, HR6) using 30 µL of tetrahydrofuran as mobile phase with 1.0 mL/min were used to complete separation. Data were collected and processed by PSS WinGPC Unity software. In this study, molecular weights (M*_n_* and M*_w_*) were calibrated against a polystyrene calibration curve. The calibration curve was generated from a retained volume obtained from polystyrene standards with many narrow molecular weights (i.e., 1.36 × 10^6^, 1.97 × 10^5^, 5.51 × 10^4^, 3.14 × 10^4^, 7.21 × 10^3^, 1.39 × 10^3^, 5.80 × 10^2^ g/mol), phenol, and acetone. The retained volume was fitted by a polynomial equation of third order and the R^2^ value of the curve fit was 0.9984.

### 2.5. Elemental Analysis of Pyrolysis Products

Elemental analysis data of pyrolysis products was obtained by Atlantic Microlab. Inc. (Norcross, GA, USA) and the sulfur, nitrogen, hydrogen, carbon contents were measured by combustion. The content of oxygen was attributed to the fact that the sum of all element contents is 1. The error of the obtained element content was 0.3% [30].

### 2.6. Structural Analysis of Pyrolysis Oil

#### 2.6.1. Quantitative ^13^C NMR

All quantitative ^13^C NMR experimental samples were derived from dissolving 100.0 mg pyrolysis oils in 450 µL (dimethyl sulfoxide)-*d*_6_ (DMSO-*d*_6_). The experimental records were made by a Bruker Avance/DMX 400 MHz, which employed a 90° pulse angle, inverse gated decoupling pulse sequence, 8 s pulse delay, and 8000 scans.

#### 2.6.2. Quantitative ^31^P NMR

Firstly, 10.0 mg pyrolysis oils were dissolved in pyridine/CDCl_3_ (1.6:1 *v*/*v*). Then the 2-chloro-4,4,5,5-tetramethyl-1,3,2-dioxaphospholane (TMDP) was added to phosphorylate the OH groups. Finally, chromium acetylacetonate and cyclohexanol were added as a relaxant agent and an internal standard, respectively. After sufficient in situ derivatization, the samples were used for quantitative ^31^P NMR measurements which used a 90° pulse angle, inverse gated decoupling pulse sequence, 25 s pulse delay, and 128 scans.

## 3. Results and Discussion

### 3.1. Yields of Pyrolysis Products

The yields of the char and pyrolysis oil from raw pine bark, pine, and Douglas-Fir bark are summarized in Table 1. For the raw pine bark studied, the yields of char decreased at higher reactor temperatures, which is consistent with the conclusions of Pan et al. [33]. As the temperature increases, the reduction in char production may be due to a larger primary decomposition of wood or secondary decomposition of char at higher temperatures [34]. At 500 °C, raw pine bark yielded largest amount (29.18 wt %) of pyrolysis oil. Similarly, Yassir et al. [18] and Ranjeet et al. [21] also found that the highest yields of pyrolysis oil produced from waste biomass are obtained around 500 °C. Therefore, in this study, pine and Douglas-Fir bark were pyrolyzed at 500 °C. The outputs of the pyrolysis oil obtained from pine and Douglas-Fir bark at 500 °C were 26.67% and 26.65%, respectively. Compared to the raw pine bark, pine and Douglas-Fir bark produced relatively less pyrolysis oils under the same condition. The higher yields of pyrolysis oil are due to the higher carbohydrate content in raw pine bark (contains ~25 wt % of pine wood), which can produce more pyrolysis oils than lignin and tannin [27,35]. Table 2 and Table 3 show HHVs, energy densification ratios, mass and energy yields of pyrolysis oil and char. They point out that the HHVs of pyrolysis oil from bark are lower than lignin pyrolysis oils (~30 MJ/kg) [27]. In contrast to the bio-oils produced by wood and agricultural residues with HHVs of approximately 20 MJ/kg [36], the pyrolysis oils from pine and Douglas-Fir bark have much higher HHVs. Furthermore, the HHVs of char are comparable with many commercial coals, such as Jefferson, Pratl-mvb coal (34.59 MJ/kg), and Northumberland No.8-Anth. coal (32.86 MJ/kg), and apparently exceed the HHVs of Green Ind.No.3-hvBb Coal (27.36 MJ/kg), German Braunkohole lignite (25.10 MJ/kg), Jhanjra Bonbahal Seam coal-R-VII (24.08 MJ/kg) [37] and torrefied pine wood (from 21.22–32.34 MJ/kg) [31]. Both energy densification ratios (1.32–1.56) and energy yields (48.40–54.31%) of char were higher than pyrolysis oils (energy densification ratios: 1.13–1.19, energy yields: 30.16–34.42%), which indicates that pyrolysis char is also a valuable product for using as bio-energy.

### 3.2. Molecular Weights and Elemental Analysis of Pyrolysis oil

The M*_n_*, M*_w_*, and M*_w_*/M*_n_* (polydispersity values of pyrolysis oil) are displayed in Table 4. This analysis indicated that the M*_w_* of pyrolysis oil is from 389 to 489, while the M*_n_* is from 250 to 294. Ingram et al. [38] examined the molecular weights of pyrolysis oil produced from pine wood, pine bark and oak wood, oak bark at 450 °C and they indicated a similar weight average molecular range (from 390 to 460). In addition, as is seen from Table 4, the molecular weight of pyrolysis oil from Douglas-Fir bark is 489, which is significantly higher than the molecular weight of pine bark pyrolysis oils (430). The main reason may be due to more lignin that the Douglas-Fir bark contains [33]; the lignin content in biomass is slightly correlated with the M*_w_* of derived bio-oil [39,40].

The elemental analysis of pyrolysis oil produced from raw pine bark, pine, and Douglas-Fir bark are presented in Table 5. It indicated that there are ~60 wt % of carbon and ~30 wt % of oxygen in the pyrolysis oils. The contents of sulfur and nitrogen are very limited (less than 1.5 wt %). In order to compare the pyrolysis oils with petroleum and starting biomass, a Van Krevelen plot is exhibited in Figure 1, which shows the relations of the ratios of O/C and H/C of fossil materials, biomass, and pyrolysis oils. The higher H/C and lower O/C value are the characteristics of valuable products, and the quality improvement of pyrolysis oil compared to biomass is readily seen.

### 3.3. NMR Analysis of Pyrolysis Oil

Table 6 shows the hydroxyl groups contents of pyrolysis oil produced from raw pine bark, pine, and Douglas-Fir bark, which indicates that the most abundant types of hydroxyl groups are aliphatic OH groups, catechol, guaiacol, and *p*-hydroxy-phenyl OH groups. David et al. [43] pyrolyzed various biomass, such as loblolly pine, corn stover, poplar, white oak, and wheat straw. They found that there were also a large number of aliphatic OH groups, guaiacol, and p-hydroxy phenyl OH groups and fewer C-5 substituted/syringyl OH groups in pyrolysis oils. The large amount of aliphatic OH groups of pyrolysis oil is due to the decomposition of one of the major components in pine and Douglas-Fir bark—cellulose which will yield levoglucosan [29] as the major product during the pyrolysis, with one molecular levoglucosan containing three aliphatic OH groups. All levoglucosan was formed by the cleavage of glycosidic bonds in the cellulose structure [29]. Our previous work [27] indicated that catechol, guaiacol, and *p*-hydroxy-phenyl OH groups are the predominant hydroxyl groups in pyrolysis oils of softwood kraft lignin, which were caused by the cleavage of ether bonds in lignin during the thermal treatment. Furthermore, as the most bonding mode of softwood lignin, β–O-4 is as high as 43%–50% in content, which is much richer than other ether linkages such as γ–O-4 and α–O-4 [44]. Therefore, the catechol, guaiacol, and *p*-hydroxyphenyl OH groups may be mostly formed by the cleavage of lignin β–O-4 bond, which is consistent with the conclusion drawn by Zhu et al. [45] Compare with pine bark pyrolysis oils, it can be seen from Table 6 that Douglas-Fir bark pyrolysis oils have more catechol, guaiacol, and *p*-hydroxy-phenyl OH groups and a fewer aliphatic OH groups, which may be attributed to more lignin and tannin and less cellulose that Douglas-Fir bark contains [33].

In order to further understand the functional groups of pyrolysis oil, ^13^C NMR was employed to investigate pyrolysis oils in detail. A database of chemical shifts for various components present in the pyrolysis oils was collected [27] and the chemical shift assignments on the basis of this study are summarized in Table 7 to facilitate analysis. Meanwhile, a typical ^13^C NMR spectrum of Douglas-Fir bark pyrolysis oils is exhibited in Figure 2. Through the integral calculation of the spectra, the contents of various functional groups in different types of pyrolysis oil are also displayed in Table 7. The ^13^C NMR results showed that the contents of levoglucosan in raw pine and pine bark pyrolysis oils were 14.33% and 14.32%, respectively, which support the conclusion that the pyrolysis oils contain a large number of aliphatic OH groups. Moreover, our previous work [46] analyzed the pyrolysis oils generated by pine residues and pine bark through HSQC–NMR and found that most aromatic C–H bonds of pyrolysis oil came from lignin components. Significant amount of aromatic carbon (~40%) in pyrolysis oils is obtained from tannin and lignin components and the aromatic C–O bonds may be formed by a radical reaction between the aromatic and aliphatic hydroxyl groups [27]. The region of the ^13^C NMR spectrum from 0 to 95.8 ppm contains approximately 50% aliphatic carbon. These aliphatic carbon atoms contribute significantly to the energy content of pyrolysis oil (Table 2) [47,48,49]. In contrast with lignin pyrolysis oils [27], there are relatively larger number of aliphatic C–O bonds in raw pine and pine bark pyrolysis oils. A large proportion of aliphatic C–O bonds belong to the levoglucosan, which is considered to be the major decomposition product of cellulose during the thermal treatment. As a decomposition product of methoxy groups during the pyrolysis, the content of methyl-aromatic carbon is relatively higher than that of lignin pyrolysis oil [27]. The extra amount of methyl-aromatic carbon is due to the decomposition of tannin, which will produce 4-methylcatechol as one of the major products.

### 3.4. Pyrolysis Mechanism Analysis

The mechanism of bark in the process of pyrolysis is very complicated. Aiming at the large amount of aliphatic OH groups present in pyrolysis oils, the main reason may be due to the cleavage of glycosidic bonds in cellulose structure, which produces a large amount of levoglucosan, with one levoglucosan molecule containing three aliphatic OH groups. Unlike the aliphatic OH groups, the catechol, guaiacol, and *p*-hydroxy-phenyl OH groups are mostly derived from the cleavage of the lignin β–O-4 bond. In addition, aromatic carbon and aliphatic carbon are the two dominant parts in pyrolysis oils. According to our previous study [27], the aromatic C–O bonds may be formed by a radical reaction between the aromatic and aliphatic OH groups and a large proportion of aliphatic C–O bonds belong to the levoglucosan. Other functional groups related to carbon may be mainly produced from various functional groups of lignin. The tentative pyrolysis pathways of two other major functional groups in the bark lignin—methoxy groups and carboxyl groups—are shown in Figure 3. The thermal decomposition of methyl groups may form aromatic C–C bonds in heavy oils and methanol in light oils. The carbonyl bonds in heavy oils, acetic acid, and water in light oils are possibly generated by thermal cleavage of the carboxyl groups.

## 4. Industrial Application of Pyrolysis Oil

As a renewable fuel, pyrolysis oils still have limited commercial application, primarily because the raw oils cannot be easily mixed with current petroleum-based transportation fuels [50]. However, valuable products separated from pyrolysis oils have been widely used in different fields depending on their properties. Our previous literature [46] concluded that all identified compounds in pyrolysis oils can be divided into nine groups: Alkanes, alcohols, benzenes, acid, phenols, ketones, aldehydes, furans, and polycyclic aromatic hydrocarbons (PAHs). Phenols and phenolic derivatives in bio-oils are widely used in the fields of food, transportation, dyes, etc. [51] As a levoglucosan with content of 5.19%–14.33% in bio-oils, it has received extensive attention in pharmaceutical manufacturing, surfactants, and is likely to be applied in the fields of archaeology and environmental science [51,52]. It is worth noting that recent literature has shown that carbonyl compounds and acetic acid in bio-oils have antibacterial and insecticidal properties, respectively, which provide a good source of bio-oil refining into natural insecticides and antibacterial agents [53]. Once the fatty acids separated in bio-oils can be made into natural pesticides, the problem in the past of using pesticides to pollute the environment may be solved. In addition, according to relevant research, the water-insoluble portion of pyrolysis oil can also be used as a component of composite asphalt binder. This means that renewable biomass-derived resources will replace fossil asphalt [54]. At present, the industrial application of pyrolysis oil has attracted wide attention. Compounds isolated from pyrolysis oils are promising as environmentally friendly industrial products.

## 5. Conclusions

Pyrolysis of raw pine bark, pine, and Douglas-Fir bark was examined. The pyrolysis oil yields of raw pine bark, pine, and Douglas-Fir bark at 500 °C were 29.18%, 26.67%, and 26.65%, respectively. The HHVs (~25 MJ/kg) of pyrolysis oil are higher than bio-oils (~20 MJ/kg) pyrolyzed from wood and agricultural residues and lower than the lignin pyrolysis oils (~30 MJ/kg), and the HHVs of the char (~31 MJ/kg) are comparable with many commercial coals. Both energy densification ratios (1.32–1.56) and energy yields (48.4–54.31%) of char are higher than the pyrolysis oils (energy densification ratios: 1.13–1.19, energy yields: 30.16–34.42%), which indicated that the pyrolysis char is also a valuable product for using as bio-energy. The elemental analysis indicated that the pyrolysis oils have a higher H/C and lower O/C value which represent a more valuable energy resource than biomass. The ^31^P NMR spectra demonstrated that the most abundant hydroxyl groups of pyrolysis oil are aliphatic OH groups, catechol, guaiacol, and *p*-hydroxy-phenyl OH groups. The large amount of aliphatic OH groups of pyrolysis oil is due to the decomposition of cellulose which will yield levoglucosan as the major product during the pyrolysis. All levoglucosan are formed by the cleavage of glycosidic bonds in the cellulose structure. Significant amounts of catechol, guaiacol, and *p*-hydroxy-phenyl OH groups of pyrolysis oil are generated from lignin and tannin components in the barks. The ^13^C NMR results indicated that there are around 40% aromatic carbon formed by the decomposition of lignin and tannin components in the bark and approximately 50% aliphatic carbon which contributes significantly to the energy content of pyrolysis oil. In this research, a new comprehensive method was established to analyze pyrolysis products through various characterizations, which can promote the high-quality screening of pyrolysis oil and further industrial applications. A clear understanding of the HHVs, molecular weight, elemental composition, and functional group content of pyrolysis oil will help us better understand the challenges of pyrolysis oil, which is a prerequisite for effective applications of pyrolysis oil.

## Figures and Tables

**Figure 1 polymers-11-01387-f001:**
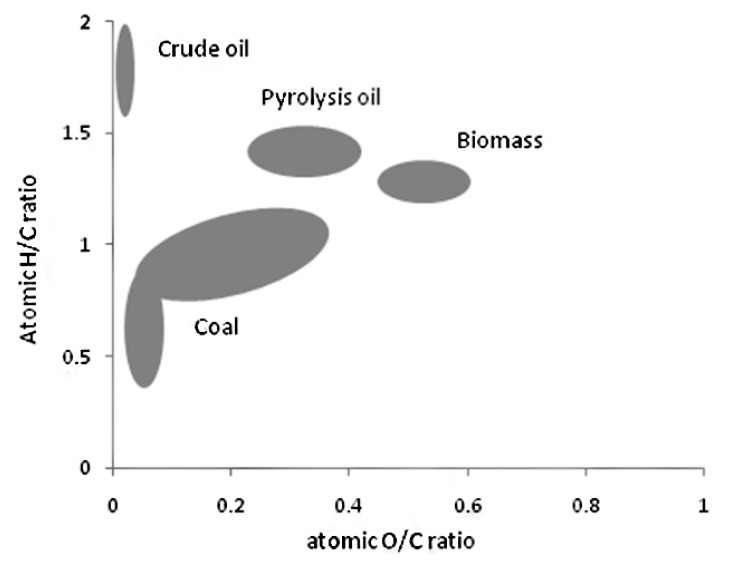
Van Krevelen figure exhibiting the atomic O/C and H/C ratios of fossil materials, biomass, and pyrolysis oils [41,42].

**Figure 2 polymers-11-01387-f002:**
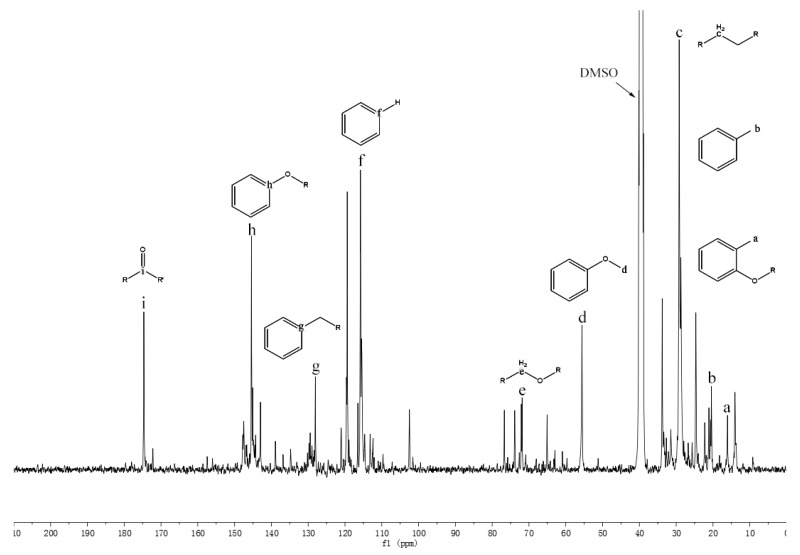
Quantitative ^13^C NMR spectrum of pyrolysis oils produced from Douglas-Fir bark at 500 °C.

**Figure 3 polymers-11-01387-f003:**
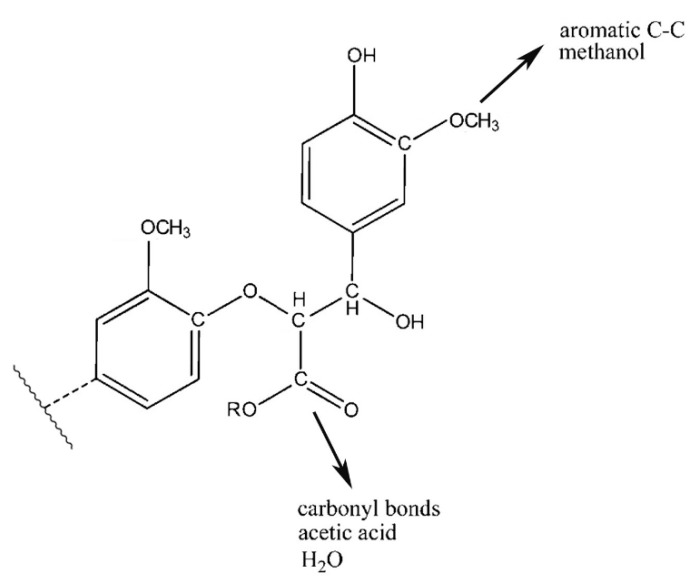
Possible pyrolysis pathway of primary decomposed functional groups in lignin during the pyrolysis.

**Table 1 polymers-11-01387-t001:** Yields of pyrolysis oil and char from raw pine bark, pine, and Douglas-Fir bark.

Softwood Bark	Temperature (°C)	Pyrolysis Oil (%)	Char ^1^ (%)
Raw pine bark	400	27.78	39.69
	500	29.18	34.74
	600	28.41	31.05
Douglas-Fir bark	500	26.65	37.55
Pine bark	500	26.67	36.68

^1^ The content of ash in raw pine bark, pine, and Douglas-Fir bark is 1.83%, 1.25%, and 1.77%, respectively.

**Table 2 polymers-11-01387-t002:** Higher heating value (HHV), energy densification ratio, mass and energy yield of pyrolysis oil produced from raw pine bark, pine, and Douglas-Fir bark.

Softwood Bark	Temperature (°C)	HHV (MJ/kg)	Energy Densification Ratio ^4^	Mass Yield (%)	Energy Yield ^5^ (%)
Raw pine bark ^1^	400	24.57	1.15	27.78	31.84
	500	25.29	1.18	29.18	34.42
	600	24.54	1.13	28.41	32.19
Douglas-Fir ^2^ bark	500	26.70	1.13	26.65	30.16
Pine bark ^3^	500	25.67	1.19	26.67	31.61

^1^ HHV for raw pine bark is 21.44 MJ/kg; ^2^ HHV for Douglas-Fir bark is 23.59 MJ/kg; ^3^ HHV for pine bark is 21.66 MJ/kg; ^4^ Energy densification ratio = HHV for pyrolysis oil/HHV for dried bark [31]; ^5^ Energy yield = energy densification ratio × mass yield [31].

**Table 3 polymers-11-01387-t003:** HHV, energy densification ratio, mass and energy yield of pyrolysis char produced from raw pine bark, pine and Douglas-Fir bark.

Softwood Bark	Temperature (°C)	HHV (MJ/kg) ^1^	Energy Densification Ratio ^5^	Mass Yield (%)	Energy Yield ^6^ (%)
Raw pine bark ^2^	400	29.34	1.37	39.69	54.31
	500	31.61	1.47	34.74	51.22
	600	33.76	1.56	31.05	48.40
Douglas-Fir ^3^ bark	500	31.11	1.32	37.55	49.52
Pine bark ^4^	500	31.77	1.47	36.68	53.80

^1^ Heating value is based on weight of ash-free char; ^2^ HHV for raw pine bark is 21.44 MJ/kg; ^3^ HHV for Douglas-Fir bark is 23.59 MJ/kg; ^4^ HHV for pine bark is 21.66 MJ/kg; ^5^ Energy densification ratio = HHV for pyrolysis char/HHV for dried bark [31]; ^6^ Energy yield = energy densification ratio × mass yield [31].

**Table 4 polymers-11-01387-t004:** M*_n_*, M*_w_*, and M*_w_*/M*_n_* distribution of pyrolysis oil produced from raw pine bark, pine, and Douglas-Fir bark.

Softwood Bark	Temperature (°C)	M*_n_* (g/mol)	M*_w_* (g/mol)	M*_w_*/M*_n_*
Raw pine bark	400	250	389	1.56
	500	263	433	1.65
	600	261	411	1.57
Douglas-Fir bark	500	294	489	1.66
Pine bark	500	260	430	1.66

**Table 5 polymers-11-01387-t005:** Elemental analysis of pyrolysis oil produced from raw pine bark, pine and Douglas-Fir bark.

Softwood Bark	Temperature (°C)	C%	H%	N%	O%	S%
Raw pine bark	400	60.82	7.14	0.79	31.25	-
	500	61.08	6.96	0.57	31.39	-
	600	61.31	7.18	0.88	30.63	-
Douglas-Fir bark	500	67.54	8.15	1.42	22.62	0.27
Pine bark	500	61.46	7.54	0.50	29.64	0.86

**Table 6 polymers-11-01387-t006:** Hydroxyl group contents of pyrolysis oil produced from raw pine bark, pine, and Douglas-Fir bark at 500 °C.

Hydroxyl Group Contents (mmol/g Pyrolysis Oil)	Aliphatic OH	C_5_ Substituted Guaiacyl Phenolic OH	Catechol, Guaiacol and *p*-Hydroxy-phenyl OH	Acid-OH
Integration region [27] (ppm)	150.0–145.5	144.7–140.2	140.2–137.3	136.6–133.6
Raw pine bark	2.63	0.64	1.96	1.07
Douglas-Fir bark	1.66	0.64	2.73	1.47
Pine bark	3.25	0.80	2.50	1.13

**Table 7 polymers-11-01387-t007:** Integration results shown as the percentage of carbon of pyrolysis oils produced from raw pine bark, pine, and Douglas-Fir bark at 500 °C.

	Integration Region (ppm)	Structure	Raw Pine Bark	Douglas-Fir Bark	Pine Bark
Carbonyl	215.0–166.5	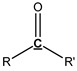	6.53	3.84	5.89
Aromatic C–O	166.5–142.0	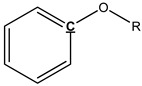	12.67	12.00	8.87
Aromatic C–C	142.0–125.0	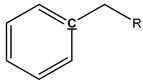	7.77	4.67	7.14
Aromatic C–H	125.0–95.8	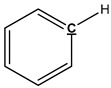	20.79	22.33	22.71
Levoglucosan	C_1_ 102.3, C_2_ 72.0, C_3_ 73.7, C_4_ 71.7, C_5_ 76.5, C_6_ 64.9	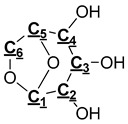	14.33	5.19	14.32
Aliphatic C–O	95.8–60.8	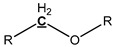	21.41	5.98	15.79
Methoxyl	60.8–55.2	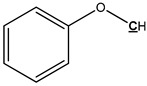	3.78	4.22	3.07
Aliphatic C–C	55.2–0.0	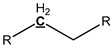	27.05	46.97	36.53
Methyl-Aromatic	21.6–19.1	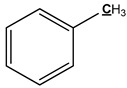	5.09	3.60	5.86
Methyl-Aromatic at ortho position of a hydroxyl or methoxyl group	16.1–15.4	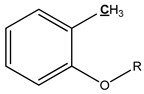	0.69	1.14	0.65

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
