# Peer review of "A Comprehensive Characterization of Pyrolysis Oil from Softwood Barks"

_polymers, 2019, doi:10.3390/polym11091387_

Round 1

Reviewer 1 Report

The authors reported a manuscript on characterization of pyrolysis products from softwood barks. This investigation is not new and has been done several times with different raw materials. However, the results might be of interest for industrial application. The publishing is suggested after below revisions:

The manuscript typing is in very low quality! :

Extra ‘’and’’ in the abstract line 16

The authors used sometime Kelvin sometime centigrade, lines 65-67

Gel permeation chromatography (GPC) should be all first letter capital!

line 85, punctuation mark!

line 103, capital letter!

129, Dates!

I would suggest that authors check the manuscript for spelling! For capital letters!

It would be more valuable if the authors include a section on the industrial application of  these pyrolysis oils

I would suggest citing the recent work of Weckhuysen et al. on pyrolysis bio-oils, DOI: 10.1021/acssuschemeng.6b01329

Reviewer 2 Report

Ben et al., was examined the Pyrolysis of raw pine bark, pine and Douglas-Fir bark. Work is really well presented. But still needs revision before publication

Title should be more attractive. A tentative mechanism regarding pyrolysis should provide. Please check it. What about the graphical abstract? For better clarification, author should provide the digital photographs of before and after pyrolysis of raw pine bark, pine and Douglas-Fir bark. Check it. Introduction should be more update by 2018 and 2019. Better provide some comparison study with recent published paper. Line 194 & 195, the used formula needs references. Check it. Line 130,…… In this study, The calibration curve of polystyrene was used as reference to calibrate Mn and Mw…Check grammar. Line 232-234….Therefore, the significant amounts of catechol, guaiacol and p-hydroxy-phenyl OH groups in pyrolysis oils are due to the thermal decomposition of another two major components in the pine and Douglas-Fir bark—lignin and tannin…needs recent updates. Check it.

Round 2

Reviewer 2 Report

Accepted.

Author Response

Thank you very much for your approval of our manuscript.